## PERSPECTIVE

# Viewing life without labels under optical microscopes

Biswajoy Ghosh [1✉] & Krishna Agarwal [1✉]

Optical microscopes today have pushed the limits of speed, quality, and observable space in biological specimens revolutionizing how we view life today. Further, specific labeling of samples for imaging has provided insight into how life functions. This enabled label-based microscopy to percolate and integrate into mainstream life science research. However, the use of labelfree microscopy has been mostly limited, resulting in testing for bio-application but not bio-integration. To enable bio-integration, such microscopes need to be evaluated for their timeliness to answer biological questions uniquely and establish a long-term growth prospect. The article presents key label-free optical microscopes and discusses their integrative potential in life science research for the unperturbed analysis of biological samples.

Historically, optical microscopy has advanced parallelly with life sciences. Today, any standard biological research facility is equipped with a microscope to image morphologies in brightfield mode and molecular distributions in epifluorescence mode to observe labeled structures of interest. This setup is a biologist's sweet spot as most studies can be performed with such a system or designed to be accommodated in them. The need for molecular quantification and precise optical sectioning made laser scanning confocal microscopy a favorite among biologists. The growing interest in fast and live imaging of thick samples in 3D has served as timely feedback to develop imaging tools like multiphoton[1–6] and light-sheet microscopes[7–9]. The need for observing cellular dynamic changes encouraged fluorescence-based methods like fluorescence resonance energy transfer (FRET)[10, 11], fluorescence lifetime imaging microscopy (FLIM)[12,13], and fluorescence recovery after photobleaching (FRAP)[14,15]. The importance of observing minute details motivated super-resolution methods like structured illumination (SIM)[16,17], stimulated emission depletion (STED)[18,19], and single-molecule localization (SMLM)[20–22] microscopy. Supporting systems like faster cameras, high-throughput adaptive automation, and innovative dyes have further widened bio-applications greatly. Still, there exists a gap where label-based methods are limited, that is in the chemically unperturbed evaluation of biological samples, without any intervention of labels and associated chemicals.

Many well-known labelfree imaging methods despite their advantages are limited when the goal is mechanistic studies or knowledge discovery. For example, brightfield, phase contrast, and differential interference contrast (DIC) microscopy are biologists' common choices. Brightfield microscopy is best suited when the samples are stained with a dye that confers the contrast. However, for live studies, getting high-contrast images for near-transparent living cells is challenging. Phase contrast and DIC imaging optically heighten the difference between the sample and the background to generate a high-contrast image and are routinely used with fluorescence microscopy. Both phase contrast and DIC imaging are very useful tools for a biologist. But they cannot quantify phase changes, as the intensity values are non-linearly related to the phase information and hence cannot be traced back to the actual changes in the sample.

Today, advancement in labelfree imaging has enabled high-resolution and high-speed imaging of morphology, dynamics, functionality, material exchange, pathogen interaction, biochemistry, and biomechanics (Fig. 1). This article discusses the wide selection of different labelfree optical

[1] UiT - The Arctic University of Norway, Tromsø, Norway. ✉email: biswajoy.ghosh@uit.no; krishna.agarwal@uit.no

microscopes, their biological applications, and their potential to grow into a powerful tool of mainstream biomedical research. This article, therefore, aims to establish grounds for choosing suitable labelfree optical microscopy tools to augment established methods and eventually play a more integrative role in knowledge discovery.

**Labelfree structural imaging**. The need for quantifying labelfree phase images motivated quantitative phase imaging methods which have majorly remained untapped in life science research. *Quantitative phase microscopy* (QPM) is meant to measure optical path delays (or phase changes) caused by the sample. Being label-free, QPM allows live imaging without chemical toxicity or signal loss due to external factors such as photo-bleaching in fluorescence microscopy. Several QPM methods today are built upon the initial theory of image formation by Ernst Abbe in 1873[23] which established an image as a

complicated interferogram formed by the superposition of light waves coming through the sample.

QPM needs to be evaluated for information fidelity and biological knowledge value to stand the test of time. The most important aspect of a QPM is handling noise to establish its quantifiability. This is governed by the spatial and temporal phase sensitivity of the system. Noise mitigation is achieved in many ways including the use of diffusers, white light illumination, and low temporal coherence light sources which improves the image quality. To reduce noise due to scanning actions, full-field QPMs emerged which are non-scanning techniques. Full-field QPMs use the interference of the light from the two planes (sample and reference) to provide information about the optical path delays introduced by the sample. The interferometric geometries determine whether the system will have a high spatial resolution like in the phase-shifting QPM or a high temporal resolution of an off-axis QPM[24]. Further, non-interferometry QPMs like the *transport of intensity* method obtain phase information without

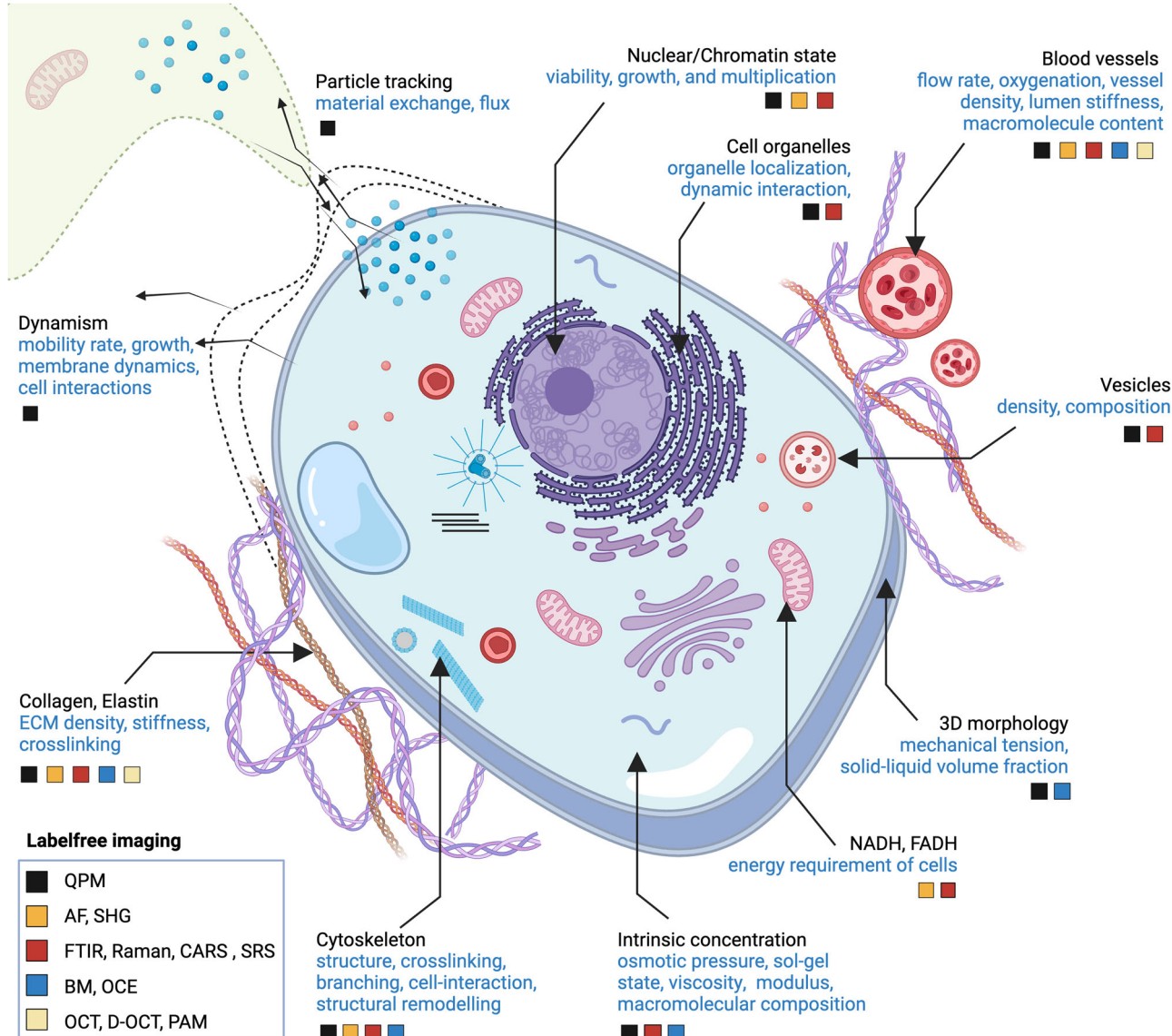

**Fig. 1 Structure is function: an overview of bio-application potential of labelfree optical microscopes.** The figure illustrates the applicability of available labelfree optical microscopes to evaluate structural, biomechanical, and biochemical attributes of the cells and tissues as a measure of their functions. QPM quantitative phase imaging, AF autofluorescence microscopy, SHG second harmonic generation microscopy, FTIR FTIR microspectroscopy, Raman Raman microspectroscopy, BM Brillouin microscopy, OCE optical coherence elastography, OCT optical coherence tomography, D-OCT Doppler-OCT, PAM photoacoustic microscopy.

specialized interferometric geometry but only two intensity images, one in focus and one slightly out of focus[25]. Although the method is easy to implement without needing very specialized systems, it is meant for low-resolution imaging. Technologies such as Fourier phase microscopy (FPM)[26], Digital holographic microscopy (DHM)[27], diffraction phase microscopy (DPM)[28], optical diffraction tomography (ODT)[29], spatial light interference microscopy (SLIM)[30], wide-field digital interferometry (WFDI)[31] are some of the prominent QPM technologies that have come up over the years. The focus of QPM today is shifting from technology development to practical applications. Today QPM technologies have demonstrated application in developmental biology[32], neuroscience[33–35], microbiology[36], pathology[37], cancer[38], genetic diseases[31], immunology[39], pharmacology[39], wound healing[36], and metabolic disorders[40]. However, the applicability of a method to any biological field needs to be parameterized to ensure the universal appeal of the method.

Broadly, QPM measures morphology, dynamism, and volume information in cells. Morphological features can be measured to determine growth, viability, response to external stimuli, or pathology using phase shift values. But the quantified phase value is combined with height and refractive index. Thus, if it is crucial to determine either of the two, they need to be correctly decoupled. The decoupling is indeed relevant as it can enable measuring additional parameters such as volume and cell mass. One decoupling method includes the use of two different refractive index mediums sequentially and measuring phase delays[41]. Other ways include the use of multiple illumination angles resulting in tomographic imaging and the use of dual wavelengths in highly dispersive media. Combining QPM with a channeled chip (milli/microfluidic channels), can measure intracellular osmolarity, cell volume changes, macromolecular concentration, shock stimuli response, and the temporal flux of molecular transport in cells[41]. Further, time-resolved QPM data has been used to measure the particle diffusion by dispersion-relation phase spectroscopy (DPS)[42]. The technique relies on intensity fluctuation due to scattered signals from the sample at a fixed angle to determine particle flow rate and predict transport characteristics (active/passive).

Phase is also innovatively used to image cell adhesion on glass surfaces and serve as a labelfree analog of total internal reflection microscopy (TIRF) by a method called interference reflection microscopy (IRM)[43, 44]. The method produces image contrast based on the phase difference between the reflected light by the glass and the cell region very close to the glass. Thus, the cell surface attached closest to the glass appears darkest due to destructive interference between the sample surface and glass reflected lights. Although the method is used for cell surface and microtubule imaging with high contrast[45], it is largely limited to a very thin region. Combining IRM with fluorescence microscopy further has also been shown to be functionally relevant in dynamic studies involving immune cells[46]. This is achieved by tracking contact points of T-cells on immunologically active cover glass, resulting in visualizing cellular contact points correlatively with fluorescently visible calcium levels.

Phase measurements by the imaging methods discussed so far are meant for thin samples, or regions close to to cover glass. This is either because they have limited range of the imaging field or are overwhelmed by strong scattering signals originating from the out-of-focus sites of the sample.

Imaging deeper in tissues, organs, and animals is a general challenge of optical microscopy. Since the 3D environment is crucial for biological functions it is important to image life in 3D. Early embryonic stages and small animals like *C. elegans*, *Drosophila*, and zebrafish larvae are now quite amenable to microscopy as they are not too dense optically. The nascent need

to image human or mammalian processes at smaller length scales directed massive interest in 3D cultures, spheroids, engineered tissues, and organoids which can easily challenge available optical methods. If resolution can be compromised, deeper regions of the sample can be accessed by using longer light wavelengths. One interferometry-based tomographic technique preceding QPMs is Optical Coherence Tomography (OCT) demonstrates spatial resolution of 5–15 μm to image up to depths of 3–5 mm. This has helped OCT's clinical integration[47] in ophthalmology, dermatology, cancer research, dentistry, gastroenterology, and cardiology. Modifications of OCT like polarization sensitive-OCT[48], OCT-angiography[49], doppler-OCT[50], and OCT-elastography[51] have enabled functional imaging of extracellular matrix, blood flow, and tissue stiffness. However, OCT like other labelfree methods suffers from noise and artifacts. Using stabilizing optical components to filter out highly scattered light and faster imaging with the Fourier domain, the sensitivity of the OCT has increased. However, the quantification especially of deep tissue regions is affected by light attenuation. To solve this, attenuation correction methods are instrumental to increase the quantifiability of OCT images in determining cancer progression[52] and wound healing[53]. Though OCT reaches deep into the samples, it cannot realize sub-micron scales where a treasure trove of biological questions lurks.

Balancing the depth and resolution is a key aspect of any 3D labelfree imaging of thick samples. For a long time, QPM did not flourish in imaging thick samples due to poor handling of multiple scattered lights that washed out details and resulted in poor image quality. However, with gradient light interference microscopy (GLIM)[32,54] phase quantification is achievable for several hundreds of microns into the sample with the ability to create tomograms enabling cross-section visualization. GLIM outshines all its predecessors in measuring embryo viability, physiological studies of 3D cultures, engineered tissues, organoids, and living organisms (like C. elegans, and zebrafish). GLIM is the label-free analog of confocal microscopy in terms of its ability to optically section the sample by suppressing the out-of-focus scattered signals. Thus, the GLIM resolution is restricted only by the optical diffraction limit.

Biological samples can not only change the light phase but also polarization. Thus, polarization light microscopy (PLM) has emerged as a good complement to phase microscopy. PLM is used to image optically anisotropic structures like fibrous proteins like collagen[55], actin[56], microtubules[57], and mitotic spindles[58] which are otherwise difficult to discern clearly in phase microscopy. PLM functions with light passing through two polarizing filters before and after the biological sample. Ideally light from the first filter cannot pass through the second one. But the anisotropy in the sample orients the light in a manner that will allow some light to pass through the second filter enabling the visualization of these anisotropic structures present in the sample. Further, the need for quantification motivated the emergence of quantitative PLM (qPLM) to measure features such as retardance and orientation[59,60]. The complementarity of the phase and polarization microscopy to measure both density and retardance fueled the combination of the two labelfree methods[61–63]. A high throughput variation of the combinatorial imaging of phase and polarization is made by integrating computational learning methods to develop quantitative label-free imaging with phase and polarization (QLIPP)[64].

Quantitative phase imaging of cellular to subcellular scales demands resolution higher than what the diffraction limit allows. This can be achieved by innovatively extracting information about the complex scattered field that encapsulates finer details of the structure being imaged before reconstruction. One method is by rotating the illumination or the sample with respect to the

other and acquiring in essence multiple perspectives of the sample. The multiple 2D images are then reconstructed using algorithms to get a 3D tomogram. This improves the lateral resolution of the QPM by a factor of two or more[34,65]. 2π-DHM – a specialized method based on DHM— uses 360° illuminations, a 405 nm blue laser UV, and two high numerical aperture (NA) oil objectives to achieve the goal[34]. The collection objective acquires light passing through the samples at different angles from 0 to 360° capturing more angular perspectives resulting in the image up to a spatial resolution of 70 nm.

Another high-contrast labelfree super-resolution imaging method is the rotating coherent scattering (ROCS) microscope[66]. ROCS can image small (150 nm) and fast structures like cellular podia, biofilaments, and virus-like nanoparticles. A fast-rotating blue laser produces a 360° illumination on the sample. The scattered light is collected in a matter of few milliseconds resulting in fast acquisition. The ability to image 100 nm particles labelfree fast has enabled an understanding of dynamic viral interactions with living cells.

In microscopy along with high spatial resolution, achieving simultaneous fast imaging is also crucial in certain biological studies. Capturing live virus-host interaction which needs spatial resolution in nanometers and temporal resolution in microseconds is thus a good fit. This is achieved in a labelfree fashion methods like coherent brightfield microscopy (COBRI)[67] and interfero-metric scattering microscopy (iSCAT)[68–70]. COBRI is a modifi-cation of the standard brightfield microscope by using spatially and temporally coherent laser light illumination. This translates to a spatial resolution of a few nanometers and a temporal resolution of 10 μs. iSCAT, on the other hand, uses the light scattered from the nanoparticles such as viruses with enhanced sensitivity using interferometry[70]. While fluorescent output is limited by the fluorophore molecule and photobleaching, in the labelfree domain, the photon output from light scattering can be increased simply by increasing the incident light (as a fixed percentage of the incident light is scattered). With a higher output, lesser time is needed to collect the same information enabling fast imaging. However, simply increasing photon intensity, besides phototoxicity, intro-duces a significant background scattering, thus defeating the purpose of ultrasensitive detection. It is here that detecting interferometric scattering comes in. In iSCAT, both the scattered light from the nanoparticles and the reflected light from the glass/ water interface is collected after tight focusing by a high NA objective. The method is ideal for smaller particles (<50 nm) as the difference between the scattered and reflected light is significantly high to create the required contrast. In addition to these methods, alternative labelfree methods are increasingly used for viral tracking today[70,71].

While phase microscopy relies on incident light being altered by the sample, biological materials are full of molecules that can be detectable by optical sensors. Photoacoustic microscopy (PAM) for example detects pigments such as hemoglobin. PAM excites the target with laser pulses, and the resulting thermal expansion of the pigments emits mechanical waves detected by the ultrasound detector. The ultrasonic detection enables PAM to image up to a depth of 1–3 mm. The lateral resolution is determined by how well the light is focused on the sample and classifies PAM as optically resolved (resolution 0.2–1 μm up to 1 mm) or acoustically resolved (resolution 2–15 μm up to 2–3 mm) variations. The lateral resolution of PAM has been pushed to 90 nm owing to innovative superresolution methods[72,73]. PAM is used in the study of blood microvasculature with clinical applications in wound asssessment[74,75], new blood vessel formation[76], diabetic foot ulcers[77], and angiogenesis[78].

Besides chromophores, our body is prevalent with autofluor-escent molecules. Autofluorescence is often a nuisance in imaging

as it interferes with the fluorescently labeled molecules. However, autofluorescence is used to unravel key pathophysiological processes like the epithelial-mesenchymal transition[79] and cancer stemness[80]. In cellular autofluorescence imaging, metabolic molecules like NADH and FADH reflect cellular energy taxation and are likely to vary in proliferation, growth, and differentiation. On the other hand, in tissues, the extracellular matrix (ECM) has collagen and elastin which indicate the tissue's mechanical integrity and is affected during remodeling, fibrosis, and cancer. Furthermore, autofluorescence can also be exploited for super-resolution microscopy to visualize nanostructures to image chromatin states in cells[81] and histopathological detection in cancer tissues[82]. The drawback of molecular specificity has limited it to disease classification purposes. Strategies including the use of tight excitation and emission filters, additional assays, structural benchmarks, and sample knowledge can be incorpo-rated to enable the utility, fidelity, and quantifiability of the method.

ECM fibrils like collagen and elastin are autofluorescent but are dense and hard to validate. Second harmonic generation imaging (SHG) is another labelfree method that detects fibril structures in biological samples and can be complementary to ECM autofluor-escence. Besides ECM, SHG can image intracellularly actomyosin complexes and microtubules. Thus it has value in evaluating disease potential[83]. SHG exemplifies the use of light modulation to bring out specific nano/microstructural information in the sample such as fibrillar form. The key advantage of SHG is its use of light polarization rather than absorption and hence reducing photo-toxicity and photobleaching. Further, as it used near-infrared light, it can be used to image thicker samples up to hundreds of microns. In addition, SHG imaging is specific to fibrillar structures and offers high structural sensitivity. The structural sensitivity and specificity make SHG translatable to several clinical applications like connective tissue diseases, fibrosis, heart and musculoskeletal conditions, and cancers[83–85]. Further, at least 2 times improvement has been achieved in SHG with several super-resolution methods[5,86,87] empowering precise quantification of fibrillar density and structural descriptions. From a technical standpoint, two-photon microscopy based has propelled much of the recent developments in autofluorescence[3,88–90] and SHG imaging[83–86,91–94] using non-linear microscopy.

## Chemical and mechanical imaging

The detection of functional changes is vital to understand biological mechanisms. Biochem-ical assays, molecular blots, and spectroscopic methods are thus tools for functional characterization globally in the sample. Given the large spatial heterogeneity in the biological samples, crucial developments often get averaged out hampering the under-standing of a missing piece of biological pathways. Hence the need for functionally imaging the sample field drove chemical staining methods and subsequent microscopic evaluations have helped biologists and pathologists greatly[95–97]. Though these methods find use in staging functional states like death, damage, organization, or multiplication, they are often hard to quantify due to subjectivities in the staining procedure. Moreover, the details are limited to the spatial resolution of the optical micro-scope, resulting in identifying changes that have already mani-fested or are at a relatively advanced stage of progression. Some fluorescence labeling can also offer chemical information through condition-bound fluorescence activation[98,99], however being limited only to a few biological functions.

To understand the early onset of diseases, early chemical functional group changes (length scales of 100–200 pm) need to be identified which are well beyond the power of conventional optical microscopes including superresolution (Fig. 2). Advances

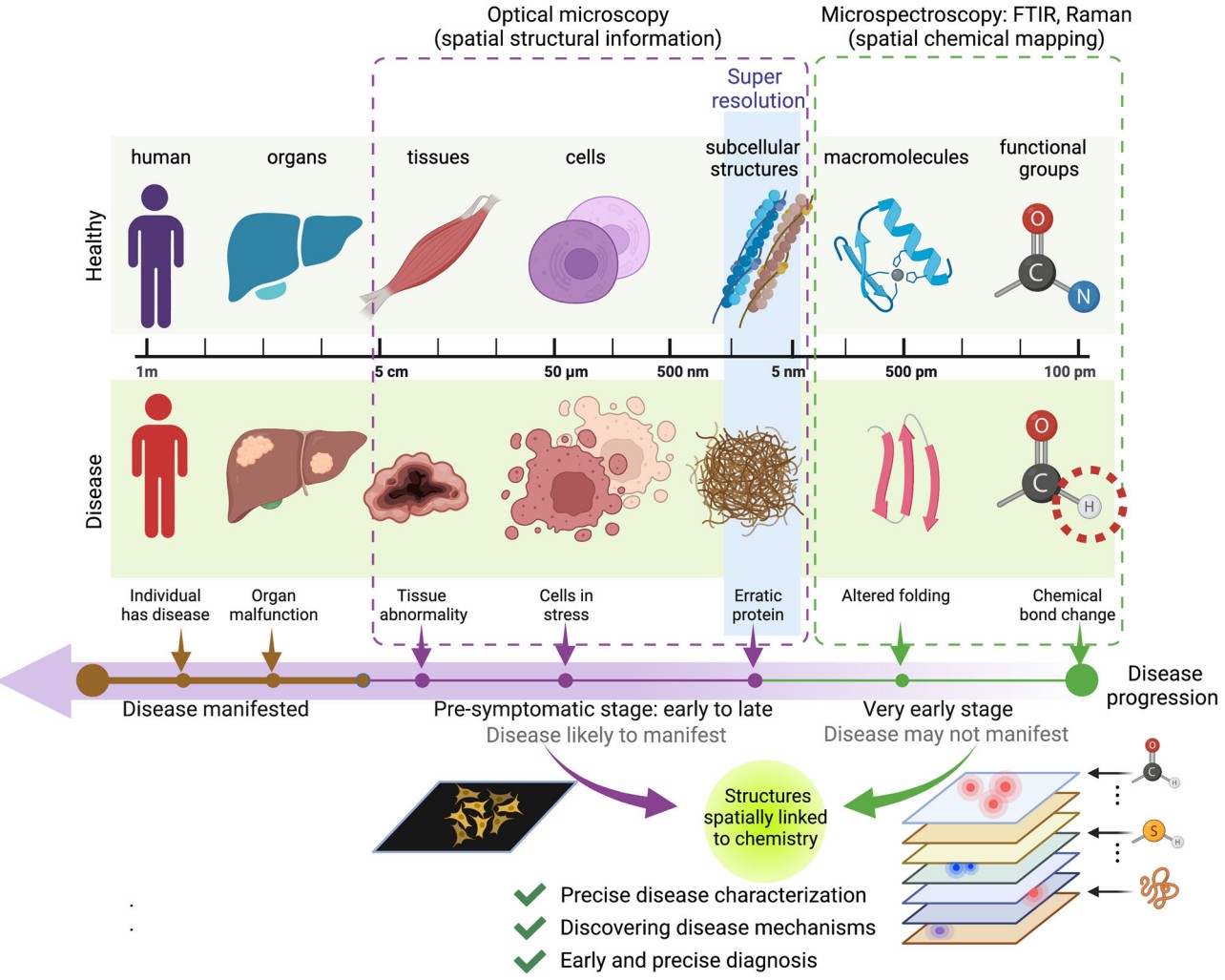

**Fig. 2 Combining structure and chemistry.** Optical microscopy and spectroscopy detect different scales of organization and hence are complementary to visualize different phases of development and disease. Optical microscopy even with the super-resolution cannot go beyond 1 nm. Microspectroscopy detects chemical changes occurring at levels of functional groups with the spatial distribution. However, with thousands of chemical changes occurring, it is difficult to use spectroscopy alone, especially at early onset, until it is correlated with manifested structural changes occurring at micron to nano scales. This can help characterize the early onset of disease as well as understand the basis of several physiological and pathological events.

in optical spectroscopy methods like the Fourier transform infrared (FTIR) spectroscopy[100] and Raman spectroscopy[101] fill this void by providing information on changes in chemical bonds and functional groups. A typical spectrum contains several peaks each corresponding to a specific chemical species. The height, breadth, and location/shifts provide key information on the concentration, diversity, and bond lengths/bond energies of the chemical species. The limiting factor of spectral resolution is also dealt with using spectral deconvolution algorithms that resolve broad peaks like amide bands to reveal sub-peaks of protein secondary structures[102]. This way spectroscopy methods are perfect complements of imaging and together can be useful to understand the mysteries of life in health and disease[102]. Yet another limiting factor is identifying spatially matched chemical information in the heterogeneous sample. This need has been largely addressed by spectral-microscopy or microspectroscopy versions of FTIR[103] and Raman[104–106]. The microspectroscopy methods use the same principle as their spectroscopy counterpart with the added ability of an objective lens spatially scan the sample. Thus, we have an image stack with an image for every wavelength across the spectral range of the imaging system. Microspectroscopy just like the spectroscopy counterpart suffers

from the same limitations as the spectroscopy equivalents such as noise management, the need for baseline adjustments, mitigating autofluorescence (for Raman), and interfering signals originating from substrate or water content. Nonetheless, with improved noise management and increased spectral and spatial resolution more detailed biochemical information are extractable.

Microscopy and microspectroscopy can be judiciously used correlatively to predict very early changes in biological systems (Fig. 2). For a long time, the resolution of microspectroscopy was not matched with advanced microscopes. This is because the former still collected spectral information from an area spanning several pixels. This has changed over the years when nanoscopy was brought to both FTIR[107–109] and Raman[110–113] microspectroscopy enabling high-resolution spectral and spatial resolution for point-to-point comparison with other high-resolution imaging methods. Further, growth of high-speed spectral imaging[114] and single molecule trapping[115, 116] modifications have improved precision in live cell and in vivo imaging[117].

Non-linear optical imaging by two-photon microscopes has not only empowered autofluorescence and SHG imaging but also spectral imaging methods like coherent anti-stokes Raman scattering (CARS) and stimulated Raman scattering microscopy

(SRS) for imaging proteins, lipid droplets, and nucleic acids[118,119]. CARS and SRS are both non-linear optical imaging and use multiple stages of excitation to collect chemical information from a spectral region specific to a chemical species. The C–H stretching regions and protein fingerprint regions are the most widely used. These regions being spectrally crowded results in the overlap of multiple chemical species thereby interfering in identifying contributions from a particular functional group. However spectral deconvolution can be used in certain cases to recover pure information in many cases[120,121].

Like chemical receptors on the cell surface, there are mechanoreceptors as well that can translate biophysical forces to enable gene expression and modulate biological functions. Mechanobiology has been demonstrated in stem cells[122], tumor progression[123], neurodegenerative diseases[106], developmental biology[124], regenerative biology[125], cell adhesion, and migration[126]. Many biomechanical studies on cells and tissues are performed indirectly today by labeling mechanosensitive molecules and observing their expressions and localization in fluorescence microscopy. However, directly imaging the dynamic mechanical changes of live cells and tissues labelfree would help understand dynamic changes in health and disease progression.

The two most popular mechanical imaging modalities are atomic force microscopy (AFM) and ultrasound elastography (USE). AFM is meant for extremely small scales (1 nm - few microns) of physical forces providing surface mechanical information. USE images at the organ scale with trans-body penetration and hence suitable for clinical settings. However, there is a gap in the length scale between a few microns to a few millimeters (cells and tissues) which has huge potential for biomechanical exploration.

The optical analog of USE is optical coherence elastography (OCE) which is a functional advancement of OCT. OCE uses an external source of tissue deformation which can be either contact-based or non-contact based to measure tissue displacements[51,127]. OCE can determine mechanical stiffness at depths of several millimeters in non-homogenous samples, which is ideal for biological tissues, 3D in vitro models, and small animal models. However, the limitation is still the resolution that is in tens of microns. This limits its applications where cellular or subcellular resolutions are warranted.

Brillouin microscopy is a labelfree mechanical imaging providing diffraction-limited resolution based on the principles of Brillouin light scattering[128] to measure longitudinal[129] or shear modulus[130]. Brillouin scattering is an inelastic scattering event occurring due to the interaction of photons from the light source and phonons (mechanical vibrations) from the sample. The phonons interact with the incident light exchanging energy and resulting in a population of inelastically scattered light (Brillouin scattering). These Brillouin scatters hence provide a measure of the mechanical property of the sample. Brillouin microscopy has been applied to cells and tissues today with high speed and low phototoxicity[129]. In cells, mechanical changes arise from cytoskeletal modifications, junctions of cell-cell or cell-matrix interactions, and the solid-liquid volume fraction of the cytoplasm and the membranes. In the extracellular matrix of tissues, the arrangement, cross-linking, and density of the proteins are responsible for the mechanical properties. Brillouin microscopy has been demonstrated in several biological models and diseases. It has been used in cell biology[131], developmental biology[132], estimating the metastatic potential of tumors[133], plaque deposition in Alzheimer's disease[104], and ECM stiffening in atherosclerosis[134] to name a few. One limitation of using Brillouin microscopy to measure elastic properties comes from the need for prior knowledge of the density and refractive index of the tissue. Further, the reliability of the method is not quite

straightforward on heterogeneous biological samples, especially in dynamic states. This is because of a possible mismatch between the mechanical relation times and the time in which the actual biological events occur. Although latitudes exist to improve the temporal resolution of the microscope, for now, it is crucial to keep the limitations into consideration and image structures that can be accommodated in the speed range slower than the acoustic relaxation. Also, since the mechanical changes impact chemical behavior and vice versa, correlative mechanical and chemical imaging can be key to revealing cause-effect relationships in life processes[104]. Further, the realization of the fact that mechanical activities are multifaceted and non-linear and not a simple measure of stiffness will go a long way to address biological questions correctly.

**The growth potential of labelfree optical microscopy.** Although labelfree microscopes have tremendous potential in many aspects of biological samples, some biological targets are more conducive to labelfree optical microscopy and have high relevance to biomedical research, we refer to them as super biotargets (Table. 1). Given the major improvements over the last couple of decades in labelfree optical microscopes, there is a huge scope for improvement in speed, resolution, quantitative accuracy, selectivity, and the possibility for automated analysis.

1. Resolution: Interest in superresolution microscopy has rapidly grown for bio-applications to observe smaller entities in action. Fluorescence-based superresolution methods like STORM, STED, PALM, and SR-SIM[135] meet this demand. The tremendous interest in superresolution and limitations associated with labeling motivated labelfree optical nanoscopy. Today the realm of labelfree nanoscopy exists with demonstrated bio-applications in phase imaging[34], photoacoustic imaging[73], autofluorescence[81], and SHG[6]. The promise of these would benefit further from reinforcement by resolving specificity and artifact characterization.

2. Speed: The realization of ultrafast imaging is not new. Speeding up imaging time without compromising image quality is achievable by the use of innovative optical hardware including patterned illumination, ultrafast focusing, and efficient detection[136,137]. The primary aim is to capture fast movements in the body across scales ranging from beating heart to subcellular cargo trafficking. The speed of imaging comes at the cost of efficient capturing of information. This includes the reliable inclusion of signals coming from the sample and handling the noise in an efficient way without compromising image quality.

3. Accuracy: Imaging accuracy is the fidelity of the imaging systems to dependably convey the information from the sample. Illumination engineering, sample handling, and detection systems can enhance the signal-to-noise ratio that decreases the artifacts. However often, the noise is an inherent part of the signal and may even carry crucial information which when exploited can reveal hidden information. Thus, the characterization of what is perceived as noise can be useful. Computational simulations can model the light-sample interaction and more specifically the signals that are expected to be detected by the microscopic system. The challenge of the current computational modeling today is to unwrap such convolutions to exploit the so-called noise contributing to more usable signal per acquisition leading to faster and more accurate imaging in all three spatial axes.

4. Selectivity: Spatial selectivity implies imaging precise 3D space of the sample. Improved accuracy brings more

**Table 1 Super biotargets for labelfree optical microscopy.**

| Structure | | Function | Compatibility with labelfree optical microscopes | Integrability into biomedical research |
|---|---|---|---|---|
| Extra cellular | Collagen[6, 55,84,144-147] | Most abundant protein, present in connective tissues, main component of the extracellular matrix, organ mechanical integrity, contains the amino acid sequence for cell-matrix adhesion, several isotypes found in the body *abnormality can occur in almost any organ with solid connective tissue* | *Modalities:* QPM, OCT (light scattering); polarization sensitive-OCT (birefringent); SHG (fibrillar structure); AF (autofluorescent); optical superresolution (AF, SHG etc.); Raman/FTIR microspectroscopy (triple helix protein, side-chain modifications chemistry, crosslinking chemistry); BM (strong Brillouin shift), OCE (optical response to deformation) *Challenges:* very dense - inhibiting light penetration, highly scattering - prone to noise | *Usable formats:* isolated commercial grade collagen from different animal origins, tissue sections (formalin-fixed, cryopreserved), 3D collagen matrix for in vitro cell culture, 3D engineered tissue, in vivo/ex vivo tissues *Spatial correlatability:* staining methods (e.g., van Gieson's, Mason trichrome, Mallory trichrome, Movat's pentachrome); immunolabelled for all or specific isotypes. *Application:* organ fibrosis, development, tissue repair, musculoskeletal disorders, cancers |
| | Blood/ vasculature[31, 40,148-154] | Includes blood cells and vessels, transport respiratory gases, nutrition, hormones, water, and waste material across the body, an important part of the immune system, wound repair *abnormality can affect all organs* | *Modalities:* QPM (RBC, tissue sections); OCT (tissues); AF (RBC), optical superresolution based on AF; PAM (hemoglobin pigment) Raman/FTIR microspectroscopy (chemical changes in hemoglobin), BM (vessel lining, blood cells), OCE (vessel lining stiffness response) *Challenges:* present deep into the tissue, restricting in situ imaging in larger organs and animals | *Usable formats:* blood smears, tissue sections, engineered in vitro blood vessels *Spatial correlatability:* blood cell staining methods, immunolabelling for endothelium (cells and lining matrix) *Application:* vasculogenesis, angiogenesis in development and cancer, anemia, leukemia |
| Intra cellular | Cytoskeleton[45, 56-58,131,155] | Includes actin, myosin, microtubules, and intermediate filaments; cell shape and structure; cell motility, cell growth and differentiation, both cell-cell and cell-matrix interaction, cell division, mechano-sensation from membrane to nucleus *abnormality can affect cell structure and function* | *Modalities:* QPM, superresolution (QPM based), SHG (fibrillar structure of actomyosin, microtubules), Raman/FTIR microspectroscopy (branching, crosslinking), BM (good Brillouin shift), *Challenges:* difficult to differentiate from other intracellular structures in labelfree methods | *Usable formats:* commercially available isolated, in vitro cell culture, tissue sections. *Spatial correlatability:* direct and indirect fluorescence labeling *application:* cell biology, fibrosis, cancer, motility, growth, cell stiffness. |
| | Nucleus[32, 81,156-158] | Includes nuclear membrane, nucleoplasm, nucleolus, and chromatin material, in all adult eukaryotic cells except RBCs, multiple nuclei in some cells like muscles, cell division, genetic information, repair, replication, transcription *abnormality can affect cell multiplication and genetic propagation* | *Modalities:* QPM, AF, AF superresolution, Raman/FTIR microspectroscopy, BM *challenges:* no major challenges | *Usable formats:* in vitro cells, 3D cell-laden engineered tissues, tissue sections *Spatial correlatability:* nuclear stains (whole nucleus), chromatin stains (e.g., Giemsa), membrane stains (nuclear membrane) *Application:* genetic diseases, cancers, cell biology, development, differentiation |

Some biological targets have tremendous potential due to their versatility and amenability to labelfree microscopy as well as their practical impact on biological and medical research. Here are enlisted some of the super biotargets and their potential for labelfree microscopy.

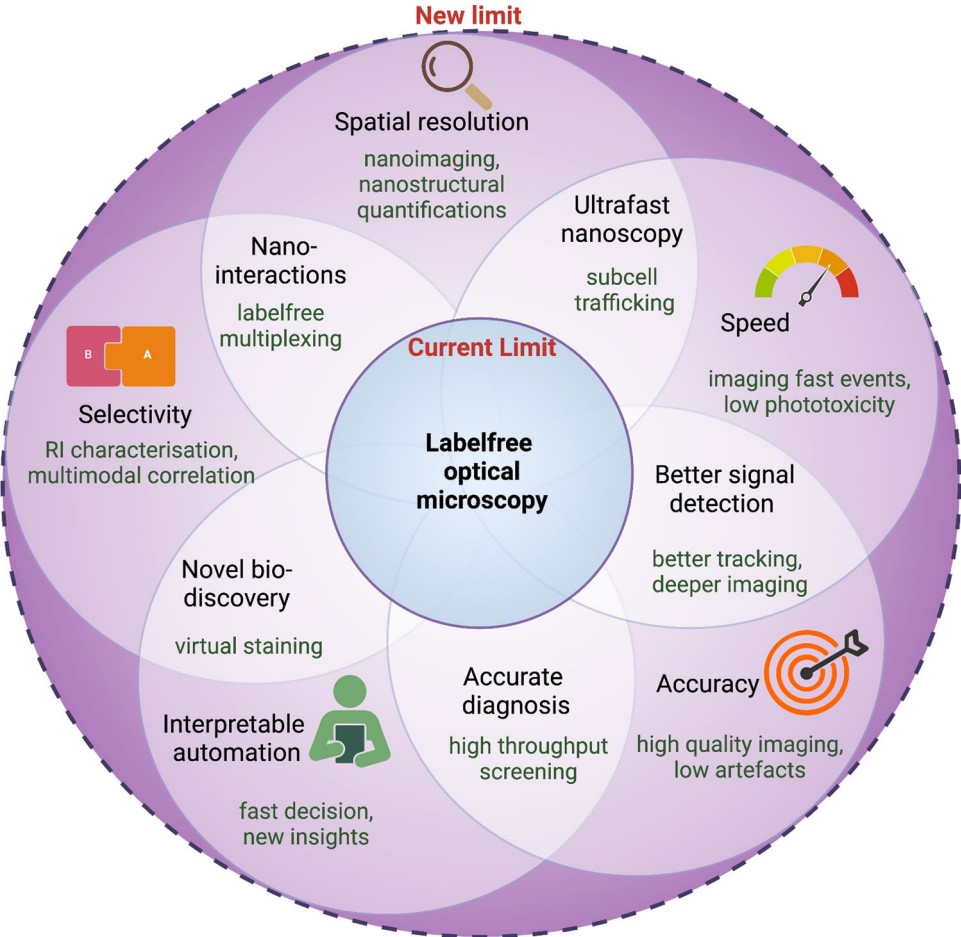

**Fig. 3 Pushing the limits of labelfree microscopy.** The figure illustrates the technological growth potential of labelfree optical microscopes. The central circle shows the current limit of labelfree optical microscopy, and the petals represent the different aspects where it can grow and eventually enlarge the scope of labelfree microscopes for biological imaging. The actual growth potential however heavily lies in the integration of the techniques into standard biomedical routines.

selective information while rejecting signals from elsewhere. Of course, this is key to integrating labelfree optical imaging into mainstream biological research. It is critical in medical diagnosis where the occurrence of a specific component is a disease marker. Selectivity can be approached in many ways. The most common is comparing known benchmarks with standards. Although it is often found the exact requirements for standards do not suit a new method and are made to fit with existing standards. The other way is to find inherent identifying factors like shape or texture. For example, in the cells' phase map, both mitochondria and actin filaments are seen, and QPM can be used to distinguish them by virtual multiplexing using a measure like the refractive index. In autofluorescence imaging, the inherent measures can be simply the excitation and emission filters to allow imaging of specific molecules.

5. Automation: Computational tools can be employed to evaluate, identify, learn, and inform the end users about structures that are otherwise indistinguishable. This can be a high-volume task during its development, but this virtual labeling of the specific structures can be rewarding in the long run for high throughput services such as clinical screening and assistance in diagnosis. Deep learning is a powerful tool to achieve tasks like classification between test groups. However, it is important to realize the basis of computational decisions. Currently of interest to

computational scientists working on interpretable neural networks with potential in knowledge mining and fundamental discoveries.

Thus, labelfree optical microscopy has come a long way in pushing the limits for multi-faceted unperturbed live imaging of biological processes. But one sustained pain point is the problem of phototoxicity that exists in optical microscopy techniques today caused by the illuminating light. It is a big problem in fluorescent imaging[138]. In the labelfree domain, phototoxicity with broadband illumination like QPM is relatively low[24] but still is significant in laser-based modalities like SHG, CARS, and spectral imaging (up to $GW/cm^2$ with femtosecond laser pulses)[139]. Phototoxicity introduces minute molecular to genetic changes limiting truly unperturbed imaging of life. Since phototoxicity depends on multiple factors including microscopy type, choice of wavelength, sample preparation, and sample type. Thus, protocols and systems are needed to be evolved to ensure best practices in live imaging.

**Outlook**. When it comes to bioimaging, the whole is truly greater than the sum of the parts. The five aspects of the development of a labelfree microscope (Fig. 3) will enlarge the overall sphere of more hidden details being captured. However, microscopes with better resolution, speed, and field of view, will keep emerging, but this alone may not be enough to answer fundamental questions in

biology and thereby get integrated into bioresearch. Strategies need to be undertaken to ensure integration rather than application. Combining several labelfree optical microscopies can be benefited from the multi-modality aspect which can complement each other by filling in missing pieces of a study. Also, correlative imaging of labelfree and label-based methods can further empower not only bioresearch but also create new niche areas for the use of labelfree microscopy. Labeling and fluorescence microscopy techniques are time-tested and are not only workhorses for high precision and high throughput outcomes but also the primary drivers for bringing microscopy closer to life scientists by creating milestones that have shaped our understanding of life today. Developments of probes like endogenous proteins and live cell-friendly dyes accommodate live imaging as well. However, a complementarity exists between labeling-based microscopes and labelfree microscopes. Understanding the two-way complementarity is understanding the mysteries of life better. Labels help us with specificity confidence in the sea of thousands of molecules. Labelfree microscopes provide the biological features as a whole and thus are replete with information. Thus, changes in a system can be often observed with such methods and help in hypothesis formulations. The label-based methods can help mine key outputs from this sea of information using specific labels to validate or invalidate such hypotheses. Thus, studies must work with a handshake between the two microscopies to maximize the output of life science research.

The development of more adaptor tools that act as bridges between biologists and labelfree microscope developers need to emerge to make labelfree microscopes indispensable to biologists. A fine example of such an adaptor tool in fluorescent labeling is the discovery of the fluorescence protein—GFP which brought a revolutionary change in bioresearch through fluorescence microscopes. Similarly, in the labelfree domain, a potential adaptor tool can be calibration and quantification phantoms made by material engineering to develop well-defined targets for refractive index, scattering, and absorbance coefficients[140]. Vibrational tags[141] for chemical bond calibration in spectral imaging such as Raman, CARS, and SRS are also good examples for chemical imaging. Tissue-mimicking phantoms[142] with defined mechanical properties can enable microscopic elastography. This integration of imaging research with research in chemistry and material science to develop standardization and benchmarking tools for labelfree microscopes can have a long-standing impact in bringing these microscopes into the mainstream.

Another challenge in bringing labelfree imaging to mainstream biological research is the need for an active effort to create value in life sciences. This means identifying common grounds with existing time-tested technologies and finding innovative ways to use labelfree imaging to answer biological questions. Many labelfree methods like IRM and iSCAT can simply be added to existing confocal microscope setups with a few modifications. More spinoff companies are developing labelfree quantitative phase microscopes and creating commercial availability with user-friendly interfaces such as phi-optics Inc and nanolive[143] microscopes. Two-photon-based non-linear microscopy methods such as SHG and CARS have commercial availability with long-standing microscope companies such as Zeiss and Leica. Another approach is a long-term collaboration between microscope developers and biological facilities to establish working systems at the experiment site and work closely with each other to refine both method and biological value.

The benefit of labelfree microscopes lies in the diversity of biological aspects they can monitor ranging from light modulation, energy exchange, chemistry, and mechanics. All these properties are inherent to the biological materials and hence are

information in their native states devoid of external perturbations to solve the burning questions that exist in biology today. Thus policymakers, funding agencies, investors, and industry need to tap into the potential of labelfree microscopy with bioscience and motivate a large workforce dedicated to achieving such activities in the future.

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

## Acknowledgements

The figures were created in BioRender.com. We thank the editors and the reviewers for their valuable feedback.

## Author contributions

B.G. and K.A. conceptualized and reviewed the article. BG wrote the paper and designed the figures.

## Funding

and Emerging Technologies Project id 964800.

## Competing interests

The authors declare no competing interests.
