## [Peer Review File · Communications Biology]

Reviewers' comments:

Reviewer #1 (Remarks to the Author):

The authors present a well described state of the art methods for label-free microscopy. The review is exhaustive and provide a good summary of the current methods, applications and challenges. Below, I give suggestions on how the authors could improve the review.

1. At the beginning of the review the authors should add further references to the methods they refer to eg. DIC, QPM, SIM, FRAP, FRET..
2. the authors state"..This is because of a possible mismatch between the mechanical relation times and the time in which the actual biological events occur..." Authors could expand, another limitation of using Brillouin to measure elastic properties comes from the need to know density and refractive index of the tissue.
3. the authors could add to table 1 an extra tab with references and/or a relevant example of the application.
4. One could also consider to add QLIPP, iSCAT techniques
5. the authors state"Raman microspectroscopy enabling high resolution spectra.. modifications have improved precision in live cell imaging" as well as full embryos by cRSI. Raman has been used in vivo to image zebrafish embryos live for anatomical tissue characterization (Stevens lab, Nat. Commun. 2020)
6. Briefly expand on the outlook vision of correlative methods and label-free & what could be the potential "adaptor tools" that act as bridges between biologists and label-free microscope developers.
7. figure 3 could be improved for clarity.

Reviewer #2 (Remarks to the Author):

The perspective article that the authors have put forward is a concise and informative piece of literature, that usefully summarizes advances made in some fields of label-free optical microscopy that the authors are most knowledgeable in.

The authors make the case for label-free optical microscopy, and highlight its potential aptly. I believe that this publication definitely fits the scope of being a perspective article, which, according to the content types guide of Communications Biology, is meant to "discuss model and ideas from a personal viewpoint". As such, I understand that this publication is supposed to contain personal opinions from experts in the field, that may have a different picture than this reviewer. Nevertheless, I feel that the authors may nevertheless benefit from some general comments to inform their viewpoint and stimulate a broader exchange of ideas with the optical microscopy community.

As such, I would propose that the author address the following, possibly integrating additional content in the perspective, in accordance to other editorial policies, or at least offer their standpoint on why they should be excluded from the article.

- 1) In my opinion, a glaring omission from this perspective article concerns Interference Reflection Microscopy (IRM), and its related, more recently developed techniques of Interferometric Scattering Microscopy (ISCAT) and Coherent Brightfield (COBRI) microscopy. While lacking the quantitative information that can be extracted from the comparable QPM methods, they have been demonstrated to be perfectly capable of distinguishing structures and even performing dynamic imaging on single cells. Therefore, not even mentioning these techniques in the present publication appears as a disservice at best to a growing microscopy community. It would be therefore be fair to acknowledge works done in ISCAT and COBRI on cellular imaging with the inclusion of at least a paragraph with appropriate references, wherever the authors feel it is most appropriate;
- 2) The authors present label free imaging in a way that, as I read it, suggests almost an antagonism with fluorescence-based methods. I am referring here especially to the very first paragraph of the

work. This is a view that I have found supported in several occasions. However, I think that presenting the methods as complementary, rather than reinforcing a strong divide between them, would be more stimulating and open new ways of approaching the discipline.

3) There is no reference, not even to topical reviews, on the topics of fluorescence microscopy in the first paragraph. Inclusion of relevant texts will only make the point stronger for the authors, and so I would urge them to fill this gap accordingly.

4) Throughout the text, I felt that the topic of phototoxicity is not sufficiently addressed. It is necessary to address this common concern in the biology community to label-free techniques, which often rely on high light doses to produce contrast, and at short wavelengths as well.

5) Although very outside of the topic, I feel this work makes a great injustice to the field of superresolution microscopy despite, paraphrasing the authors, spurring the development of nanoscopic label-free methods. They are only mentioned in a hastily manner, and dismissed as suffering from "photobleaching, phototoxicity, and difficult-to-characterize artifacts". Hardly the treatment that highly appreciated and established techniques deserve! I advise the authors against these kinds of fast remarks unless they intend to embark on a full analysis, which I imagine is beyond the scope of this perspective.

6) It would be informative and definitely useful if the authors included some comments on the commercial availability of most of the techniques mentioned. Of course I do not mean including specific products or manufacturers, but rather advising the community whether or not specific technical competences are necessary to implement the suggested techniques in a microscopy facility or a specific laboratory. This is often the highest hurdle for the widespread adoption of specific techniques, and certainly a forward-looking perspective should include a mention on the topic. I look forward to receiving a response to these comments, which I hope the authors will receive positively and appropriately address.

Response to reviewers' comments

Reviewer #1 (Remarks to the Author):

The authors present a well described state of the art methods for label-free microscopy. The review is exhaustive and provide a good summary of the current methods, applications and challenges. Below, I give suggestions on how the authors could improve the review.

Response: We are grateful to the reviewer for their positive feedback and valuable comments to improve on the manuscript. The reviewer has identified some key areas like the inclusion of techniques like QLIPP and iSCAT as well as the expansion of certain concepts in the outlook and figure elements which have improved the manuscript for completeness. Below are the point-wise response to your comments. We have highlighted the changes in red in the manuscript. We hope that the reviewer is satisfied with the revisions we have made.

1. At the beginning of the review the authors should add further references to the methods they refer to eg. DIC, QPM, SIM, FRAP, FRET.

Response: We have added references to these techniques in the first paragraph of the manuscript.

2. the authors state "This is because of a possible mismatch between the mechanical relation times and the time in which the actual biological events occur..." Authors could expand, another limitation of using Brillouin to measure elastic properties comes from the need to know density and refractive index of the tissue.

Response: Thanks for pointing this out. We have included this now on page 7 highlighted in red.

3. the authors could add to table 1 an extra tab with references and/or a relevant example of the application.

Response: Thanks for the suggestions. We have added the references in the first column of the table for relevant examples of application in Table 1.

4. One could also consider to add QLIPP, iSCAT techniques

Response: We have added both QLIPP and iSCAT techniques in the revised manuscript in page 3(last para) and 4 (third para) highlighted in red.

5. the authors state "Raman microspectroscopy enabling high-resolution spectra.. modifications have improved precision in live cell imaging" as well as full embryos by cRSI. Raman has been used in vivo to image zebrafish embryos live for anatomical tissue characterization (Stevens lab, Nat. Commun. 2020)

Response: Thanks for the reference. We have included this now in the revised manuscript- page 6.

6. Briefly expand on the outlook vision of correlative methods and label-free & what could be the potential "adaptor tools" that act as bridges between biologists and label-free microscope developers.

Response: we have expanded upon the adaptor tools in the outlook section now highlighted in red.

7. figure 3 could be improved for clarity.

Response: We have now modified the figure for clarity.
Thank you for the constructive review of our work. Hope you find the revisions satisfactory.

Reviewer #2 (Remarks to the Author):

The perspective article that the authors have put forward is a concise and informative piece of literature, that usefully summarizes advances made in some fields of label-free optical microscopy that the authors are most knowledgeable in.

The authors make the case for label-free optical microscopy and highlight its potential aptly. I believe that this publication definitely fits the scope of being a perspective article, which, according to the content types guide of Communications Biology, is meant to “discuss model and ideas from a personal viewpoint”. As such, I understand that this publication is supposed to contain personal opinions from experts in the field, that may have a different picture than this reviewer. Nevertheless, I feel that the authors may nevertheless benefit from some general comments to inform their viewpoint and stimulate a broader exchange of ideas with the optical microscopy community.

As such, I would propose that the author address the following, possibly integrating additional content in the perspective, in accordance to other editorial policies, or at least offer their standpoint on why they should be excluded from the article.

Response: We are very thankful to the reviewer for the positive feedback and the valuable discussion points were constructive. The comments from the reviewer have been very insightful as we could clearly understand the reasoning behind the need for some inclusions suggested. We are happy to incorporate the suggestions in a way it fits into the intended narrative. We are content with the resulting improvement in the manuscript. We hope that the reviewer is satisfied with the revisions made.

1) In my opinion, a glaring omission from this perspective article concerns Interference Reflection Microscopy (IRM), and its related, more recently developed techniques of Interferometric Scattering Microscopy (ISCAT) and Coherent Brightfield (COBRI) microscopy. While lacking the quantitative information that can be extracted from the comparable QPM methods, they have been demonstrated to be perfectly capable of distinguishing structures and even performing dynamic imaging on single cells. Therefore, not even mentioning these techniques in the present publication appears as a disservice at best to a growing microscopy community. It would be therefore be fair to acknowledge works done in ISCAT and COBRI on cellular imaging with the inclusion of at least a paragraph with appropriate references, wherever the authors feel it is most appropriate;

Response: We have incorporated IRM (Page 2, 3rd para), iSCAT, and COBRI (page 4, 3rd para) in the revised manuscript highlighted in red

2) The authors present label free imaging in a way that, as I read it, suggests almost an antagonism with fluorescence-based methods. I am referring here especially to the very first paragraph of the work. This is a view that I have found supported in several occasions. However, I think that presenting the methods as complementary, rather than reinforcing a strong divide between them, would be more stimulating and open new ways of approaching the discipline.

Response: We fully acknowledge the importance and complementarity of label-based techniques. To remove any confusion in that regard, we have added a paragraph in the outlook

section to highlight the 2-way complementarity of the two fields of microscopy highlighted in red.

3) There is no reference, not even to topical reviews, on the topics of fluorescence microscopy in the first paragraph. Inclusion of relevant texts will only make the point stronger for the authors, and so I would urge them to fill this gap accordingly.

Response: We have added the relevant references in the first paragraph now.

4) Throughout the text, I felt that the topic of phototoxicity is not sufficiently addressed. It is necessary to address this common concern in the biology community to label-free techniques, which often rely on high light doses to produce contrast, and at short wavelengths as well.

Response: We agree with the comment. We have now added a paragraph on page 9, the first paragraph highlighted in red.

5) Although very outside of the topic, I feel this work makes a great injustice to the field of superresolution microscopy despite, paraphrasing the authors, spurring the development of nanoscopic label-free methods. They are only mentioned in a hastily manner, and dismissed as suffering from “photobleaching, phototoxicity, and difficult-to-characterize artifacts”. Hardly the treatment that highly appreciated and established techniques deserve! I advise the authors against these kinds of fast remarks unless they intend to embark on a full analysis, which I imagine is beyond the scope of this perspective.

Response: We understand that the statement does not do justice. Hence, we have decided to withdraw this statement in the revised manuscript.

6) It would be informative and definitely useful if the authors included some comments on the commercial availability of most of the techniques mentioned. Of course I do not mean including specific products or manufacturers, but rather advising the community whether or not specific technical competences are necessary to implement the suggested techniques in a microscopy facility or a specific laboratory. This is often the highest hurdle for the widespread adoption of specific techniques, and certainly a forward-looking perspective should include a mention on the topic.

Response: We have added this in the **outlook section** of the revised manuscript highlighted in red.

I look forward to receiving a response to these comments, which I hope the authors will receive positively and appropriately address.

Thank you for the detailed review of our work. Hope you find the revisions satisfactory.

Reviewers' comments:

Reviewer #2 (Remarks to the Author):

The modifications that the authors have introduced to the manuscript so far are satisfactory, as well as the addition of relevant references to additional label free and microscopy techniques. I have only very minor comments

1) The mentions of Superresolution fluorescence microscopy techniques are not complete without the addition of STED microscopy, alongside SIM and SMLM. There are several reviews in literature that address this technique that can be referenced for this.

2) A valuable application of IRM in combination with fluorescence microscopy is the following, and it may be beneficial to add it as an additional reference

(<https://www.science.org/doi/pdf/10.1126/sciadv.1603032>)

3) Another LabelFree technique that should deserve more attention, and that only now I noticed was missing is ROCS microscopy (e.g. <https://www.nature.com/articles/s41467-022-29091-0>), which is extremely relevant for this review, given its capacity to reach superresolution without fluorescent labels.

4) The sentence "Another roadblock to integrating new labelfree methods is the additional effort to avail them for value creation" is very difficult to understand due to the word choice.

I look forward to receiving a further revision of the manuscript.

Response to Reviewers' Comments

Reviewer #2 (Remarks to the Author):

The modifications that the authors have introduced to the manuscript so far are satisfactory, as well as the addition of relevant references to additional label-free and microscopy techniques. I have only very minor comments

Response: We thank the reviewer for their valuable feedback on the revised manuscript. We really appreciate this engagement and discussion. Further, the references suggested are very interesting and relevant to the theme of the manuscript. We have now incorporated all the new additions indicated in the revised version. We hope that the reviewer would find this version satisfactory.

1) The mentions of Superresolution fluorescence microscopy techniques are not complete without the addition of STED microscopy, alongside SIM and SMLM. There are several reviews in literature that address this technique that can be referenced for this.

Response: We have now added STED microscopy and suitable references on Page 1, highlighted in red.

2) A valuable application of IRM in combination with fluorescence microscopy is the following, and it may be beneficial to add it as an additional reference (<https://www.science.org/doi/pdf/10.1126/sciadv.1603032>)

Response: We thank the reviewer for the interesting paper. This indeed is a relevant paper where IRM has been used to assess functional changes labelfree in T-cell podia structures correlatively with Ca²⁺ levels. We have now incorporated this reference on page 2 highlighted in red.

3) Another LabelFree technique that should deserve more attention, and that only now I noticed was missing is ROCS microscopy (e.g. <https://www.nature.com/articles/s41467-022-29091-0>), which is extremely relevant for this review, given its capacity to reach superresolution without fluorescent labels.

Response: We have now incorporated a brief description of ROCS along with this reference on page 4 highlighted in red.

4) The sentence "Another roadblock to integrating new labelfree methods is the additional effort to avail them for value creation" is very difficult to understand due to the word choice. I look forward to receiving a further revision of the manuscript.

Response: We have now modified the statement for clarity on page 9/10 highlighted in red.